# Traveling skyrmions in chiral antiferromagnets

**Stavros Komineas[1] and Nikos Papanicolaou[2]**

**1** Department of Mathematics and Applied Mathematics,
University of Crete, 71003 Heraklion, Crete, Greece
**2** Department of Physics, University of Crete, 71003 Heraklion, Crete, Greece

## Abstract

Skyrmions in antiferromagnetic (AFM) materials with the Dzyaloshinskii-Moriya (DM) interaction are expected to exist for essentially the same reasons as in DM ferromagnets (FM). It is shown that skyrmions in antiferromagnets with the DM interaction can be traveling as solitary waves with velocities up to a maximum value that depends on the DM parameter. Their configuration is found numerically. The energy and the linear momentum of an AFM skyrmion lead to a proper definition of its mass. We give the details of the energy-momentum dispersion of traveling skyrmions and explore their particle-like character based on exact relations. The skyrmion number, known to be linked to the dynamics of topological solitons in FM, is, here, unrelated to the dynamical behavior. As a result, the solitonic behavior of skyrmions in AFM is in stark contrast to the dynamical behavior of their FM counterparts.



# 1 Introduction

Topological magnetic solitons have been studied extensively for ferromagnets (FM) and weak ferromagnets. In both cases a nonvanishing magnetization develops in the ground state, albeit by a different physical mechanism, which allows a detailed experimental investigation by standard techniques [1, 2]. In contrast, direct experimental evidence for pure antiferromagnetic (AFM) solitons is rare and has only been reported in recent years [3–5]. Theoretical arguments suggest that such solitons should exist for essentially the same reasons as in ordinary FM. It is therefore expected that vortices and chiral skyrmions that have been observed and studied in FM with the Dzyaloshinskii-Moriya (DM) interaction [6] have their counterparts in AFM [7], [8]. On the other hand, the dynamics of solitons in AFM is expected to be substantially different [9].

The dynamics of the magnetic microstructure in AFM is governed by suitable extensions of the relativistic nonlinear $\sigma$-model [2, 10–12] instead of the Landau-Lifshitz equation in FM. The relevance of the $\sigma$-model for the description of antiferromagnets became apparent through standard hydrodynamic approaches [13–15]. Detailed applications to AFM solitons were carried out mostly in the Soviet literature reviewed in part in [2], and more recently in [8, 16, 17]. The type of dynamics of magnetic solitons supported by the $\sigma$-model allows for traveling solitons and it is thus very different than the dynamics in ferromagnets. A topological soliton in a FM, such as a chiral skyrmion, is characterized by a topological number, called the skyrmion number in this context. A direct link between the skyrmion number and the dynamics of topological solitons in FM [18] was already apparent in the, so-called, Thiele equation for rigid vortex motion [19]. In contrast to FM, the skyrmion number is not linked to the dynamics of AFM solitons; instead, a different topological number was shown to be linked to AFM soliton dynamics only in the case that an external magnetic field is applied [11].

The above remarks paint an intriguing picture for the dynamics of topological solitons in AFM. In this work, we will focus on AFM materials with the DM interaction, such as those studied in Ref. [8], [20] and we will study the dynamics of chiral skyrmions. The existing results leave open the possibility for driving skyrmions in AFM as ordinary Newtonian particles, without experiencing a skew deflection (or Magnus force dynamics) seen for topological solitons in FM. In fact, traveling solitons can be readily found within the standard $\sigma$-model by means of a Lorentz transformation, but the issue remains to be studied within an extension of the $\sigma$-model for chiral magnets.

We show that topological solitons, such as skyrmions, present solitary wave behavior and they can propagate with a velocity up to a maximum value that depends on the DM parameter. We calculate the details of the traveling skyrmion configuration and we show that a mass can be naturally associated to a skyrmion. The particle-like character of AFM skyrmions is shown via their dispersion relation. Although our results are obtained within the conservative $\sigma$-model the details can help guide any subsequent efforts to exploit the dynamics of skyrmions by applying external forces, such as spin-torques.

The outline of the paper is as follows. Sec. 2 introduces the discrete model for spin dynamics in AFM and a continuous theory derived from this, that is, a nonlinear $\sigma$-model. Sec. 3 presents numerical solutions for traveling AFM skyrmions within the $\sigma$-model and the discrete model and examines their features. Sec. 4 studies the energy and momentum of traveling skyrmions and examines their particle-like character. Sec. 5 contains our concluding remarks. In Appendix A we give the details of the derivation of the $\sigma$-model. In Appendix B we derive virial relations for traveling skyrmions that are used extensively in the main text. Appendix C gives some results for the Lorentz invariant model.

## 2 The nonlinear sigma model

### 2.1 The discrete model

As a model for the magnet in two dimensions, we consider a square lattice of spins $\boldsymbol{S}_{i,j}$ with a fixed length $\boldsymbol{S}_{i,j}^2 = s^2$, where $i,j$ are integer indices for the spin site. Magnetic materials with crystal structure of low symmetry present exchange interactions with both a symmetric and an antisymmetric part and the latter is usually called the Dzyaloshinskii-Moriya interaction [21, 22]. We write a discrete Hamiltonian on the square lattice and we include symmetric exchange, a DM term, and an anisotropy term,

$$E = E_{\text{ex}} + E_{\text{DM}} + E_{\text{a}}. \tag{1}$$

The symmetric part of the exchange energy is

$$E_{\text{ex}} = J \sum_{i,j} \boldsymbol{S}_{i,j} \cdot (\boldsymbol{S}_{i+1,j} + \boldsymbol{S}_{i,j+1}), \qquad J > 0, \tag{2}$$

where antiferromagnetic coupling has been assumed. For the DM interaction we are motivated by the material $K_2V_3O_8$ [23] and we will follow Refs. [8], [20]. We will use only a simplified version of the DM energy, and we will omit the part responsible for weak ferromagnetism. We set

$$E_{\text{DM}} = D \sum_{i,j} \left[ \hat{\boldsymbol{e}}_2 \cdot (\boldsymbol{S}_{i,j} \times \boldsymbol{S}_{i+1,j}) - \hat{\boldsymbol{e}}_1 \cdot (\boldsymbol{S}_{i,j} \times \boldsymbol{S}_{i,j+1}) \right], \tag{3}$$

where $\hat{\boldsymbol{e}}_i$, $i = 1, 2, 3$ denote the unit vectors in spin space. We consider an anisotropy term of the easy-axis type

$$E_{\text{a}} = -\frac{g}{2} \sum_{i,j} [(\boldsymbol{S}_{i,j})_3]^2, \tag{4}$$

where $(\boldsymbol{S}_{i,j})_3$ denotes the third component of a spin vector.

The equation of motion for the spins is derived from the Hamiltonian and reads

$$\frac{\partial \boldsymbol{S}_{i,j}}{\partial t} = \boldsymbol{S}_{i,j} \times \boldsymbol{F}_{i,j} - \tilde{\alpha} \boldsymbol{S}_{i,j} \times \frac{\partial \boldsymbol{S}_{i,j}}{\partial t},$$
$$\boldsymbol{F}_{i,j} = -\frac{\partial E}{\partial \boldsymbol{S}_{i,j}}, \tag{5}$$

where $\boldsymbol{F}$ is the effective field. The first term on the right side of the equation conserves the energy and the second one is a damping term with $\tilde{\alpha}$ the dissipation constant. The explicit form of the effective field is

$$\begin{aligned}
\boldsymbol{F}_{i,j} = {} & -J(\boldsymbol{S}_{i+1,j} + \boldsymbol{S}_{i,j+1} + \boldsymbol{S}_{i-1,j} + \boldsymbol{S}_{i,j-1}) \\
& + D \left[ \hat{\boldsymbol{e}}_2 \times (\boldsymbol{S}_{i+1,j} - \boldsymbol{S}_{i-1,j}) - \hat{\boldsymbol{e}}_1 \times (\boldsymbol{S}_{i,j+1} - \boldsymbol{S}_{i,j-1}) \right] \\
& + g(\boldsymbol{S}_{i,j})_3 \hat{\boldsymbol{e}}_3.
\end{aligned} \tag{6}$$

### 2.2 The continuum approximation

Further analysis will be greatly facilitated if we pass to a continuum model for the antiferromagnet [2, 10–12]. The derivation of this model is given in Appendix A and it is based on a tetramerization of the original spin lattice. The order parameter is the continuous Néel vector $\boldsymbol{n} = \boldsymbol{n}(x, y, \tau)$ defined in Eq. (35), with components $(n_1, n_2, n_3)$, and it satisfies the constraint $\boldsymbol{n}^2 = 1$. The space variables $x, y$ and time $\tau$ are defined in Eqs. (37) and (38) respectively. In

the conservative case ($\tilde{\alpha} = 0$), it satisfies Eq. (42), which, after suitable rescaling of the space variables (setting $\kappa = 1$), is written as

$$\boldsymbol{n} \times (\ddot{\boldsymbol{n}} - \boldsymbol{f}) = 0,$$
$$\boldsymbol{f} = \Delta \boldsymbol{n} + 2\lambda \epsilon_{\mu\nu} \hat{\boldsymbol{e}}_{\mu} \times \partial_{\nu} \boldsymbol{n} + n_3 \hat{\boldsymbol{e}}_3,$$

(7)

where the dot denotes differentiation with respect to the scaled time variable $\tau$, $\Delta$ denotes the Laplacian in two dimensions, $\epsilon_{\mu\nu}$ is the antisymmetric tensor with $\mu, \nu = 1, 2$, and the summation convention for repeated indices is adopted. The notation $\partial_1, \partial_2$ denotes differentiation with respect to $x, y$ respectively.

Model (7) is an extension of the nonlinear $\sigma$-model. It is Hamiltonian with energy (see also Refs. [7, 24])

$$E = \tfrac{1}{2} \dot{\boldsymbol{n}}^2 + V,$$
$$V = \tfrac{1}{2} (\partial_{\mu} \boldsymbol{n}) \cdot (\partial_{\mu} \boldsymbol{n}) - \lambda \epsilon_{\mu\nu} \hat{\boldsymbol{e}}_{\mu} \cdot (\partial_{\nu} \boldsymbol{n} \times \boldsymbol{n}) + \frac{1}{2} (1 - n_3^3),$$

(8)

and the effective field in Eq. (7) is derived from $\boldsymbol{f} = -\delta V / \delta \boldsymbol{n}$. The DM energy is written in terms of the, so-called, Lifshitz invariants

$$\mathcal{L}_{\mu\nu} = \hat{\boldsymbol{e}}_{\mu} \cdot (\partial_{\nu} \boldsymbol{n} \times \boldsymbol{n})$$

(9)

that will also appear in various formulae in the following. The ground state is the uniform state for $\lambda < 2/\pi$ and the spiral state for $\lambda > 2/\pi$ [25]. We will study isolated skyrmions that are excited localised states on a uniform background.

The magnetization $\boldsymbol{m}$ is defined in Eq. (35) as the mean value of the spin on the tetramers. It is an auxiliary field in this theory, given in terms of $\boldsymbol{n}$ by Eq. (40) (repeated here for completeness),

$$\boldsymbol{m} = \frac{\epsilon}{2\sqrt{2}} \boldsymbol{n} \times \dot{\boldsymbol{n}},$$

(10)

where $\epsilon$ is a small parameter introduced in the definition of the scaled space variable through Eq. (37). It appears that the magnetization goes to zero in the limit of small $\epsilon$ where Eq. (10) is valid. But, one should recall that the definition of the scaled time in Eq. (38) contains $\epsilon$ and, therefore, in the physical units used for a specific material, the value of the magnetization vector will be nonzero [10, 12].

One should notice that the static sector of the $\sigma$-model (7) for the Néel vector in an antiferromagnet is identical to the static sector of the Landau-Lifshitz equation for the magnetization vector of a ferromagnet with corresponding interactions (exchange, DM, and anisotropy). We therefore expect that the static solitons (skyrmions, vortices, etc) obtained in an AFM precisely correspond to their counterparts in a FM. On the other hand, the dynamics of these solitons is different in AFM compared to FM. This is due to the different dynamical sectors of the $\sigma$-model and the Landau-Lifshitz equation, a point elaborated upon in App. B in connection with Eqs. (48) and (49).

## 3 Traveling skyrmion profiles

If we include the standard Gilbert damping, as it appears in the discrete Eq. (5), the continuum model (7) is extended as follows,

$$\boldsymbol{n} \times (\ddot{\boldsymbol{n}} - \boldsymbol{f} + \alpha \dot{\boldsymbol{n}}) = 0,$$

(11)

where the damping constant in the discrete and in the continuous models are related by $\alpha = (\epsilon/2)\tilde{\alpha}$. We can derive a relaxation algorithm by assuming that the damping term dominates, $\alpha \to \infty$. This is equivalent to neglecting the second time derivative in Eq. (11) and setting $\alpha = 1$ (or rescaling time) thus obtaining the relaxation algorithm

$$\dot{\boldsymbol{n}} = -\boldsymbol{n} \times (\boldsymbol{n} \times \boldsymbol{f}). \tag{12}$$

For any initial configuration, the above algorithm will lead to a local minimum of the energy in the limit $t \to \infty$. Eq. (12) is identical to the relaxation algorithm used for magnetization configurations satisfying the Landau-Lifshitz equation. We have applied (12) in order to find static AFM skyrmion solutions. The result is, obviously, identical to the chiral skyrmion configurations or profiles found for FM [26,27], except that, here, the skyrmion configuration refers to the field $\boldsymbol{n}$, and, according to Eq. (10), $\boldsymbol{m} = 0$.

We are interested in skyrmions traveling as solitary waves, that is, solutions of Eq. (7) of the form

$$\boldsymbol{n} = \boldsymbol{n}(x - \upsilon\tau, y), \tag{13}$$

where $\upsilon$ is the velocity of propagation and we have chosen $x$ as the direction of propagation. It is instructive to note that such solutions would be obtained in a straightforward way if the DM interaction were not present in Eq. (7). In that case, the model would be Lorentz invariant, i.e., for any static solution $\boldsymbol{n}_0(x, y)$, a traveling solution would be obtained by applying the Lorentz transformation

$$\boldsymbol{n}(x, y, \tau; \upsilon) = \boldsymbol{n}_0(\xi, y), \qquad \xi = \frac{x - \upsilon\tau}{\sqrt{1 - \upsilon^2}}, \tag{14}$$

and the velocity of propagation can be chosen in the interval $0 \le \upsilon < 1$. Some basic results for Lorentz invariant models are reviewed in Appendix C. When the DM interaction is present, one cannot obtain traveling solutions by simply invoking the Lorentz transformation, nevertheless, we will find numerically that traveling skyrmion solutions do exist for $\lambda \neq 0$.

We insert the traveling wave form (13) in Eq. (7) and obtain

$$\boldsymbol{n} \times \left(\boldsymbol{f} - \upsilon^2 \partial_1^2 \boldsymbol{n}\right) = 0. \tag{15}$$

Solutions of the latter equation can be found by using the relaxation algorithm (12) where instead of $\boldsymbol{f}$ we have to use $\boldsymbol{f} - \upsilon^2 \partial_1^2 \boldsymbol{n}$. We apply the algorithm using as an initial condition a Belavin-Polyakov skyrmion. Dirichlet boundary conditions are applied with $\boldsymbol{n} = (0, 0, 1)$ at the lattice end points. We typically use a lattice spacing $\Delta x = 0.1$. The algorithm converges to a skyrmion that is a solution of Eq. (15), for a range of velocities $0 \le \upsilon < \upsilon_c$. For the parameter value $\lambda = 0.45$, we find a maximum velocity $\upsilon_c \simeq 0.715$.

Fig. 1 shows the configurations for the field $\boldsymbol{n}$ for a static skyrmion and for skyrmions traveling with various velocities. As the velocity increases the skyrmion gets elongated along the axis perpendicular to the direction of propagation. There also is a smaller elongation of the configuration along the axis of propagation. This is very different than the configuration of traveling solitons under the Lorentz transformation (14), where the soliton is actually just contracted along the direction of propagation. When the velocity approaches the critical velocity $\upsilon_c$ the skyrmion core expands in space, apparently to become infinitely elongated in the limit $\upsilon \to \upsilon_c$. Refs. [28,29] report numerical observations for elliptical deformation of AFM skyrmions when these are set in motion by spin-Hall torque.

The maximum velocity of propagation is $\upsilon_c < 1$, that is, it is lower for the model with DM interaction compared to the value $\upsilon_c = 1$ attained in the Lorentz invariant model (for $\lambda = 0$). We expect that $\upsilon_c \to 1$ as $\lambda \to 0$. On the other hand, $\upsilon_c$ decreases as the DM parameter $\lambda$ is approaching the value $2/\pi$ (where the skyrmion radius becomes large).

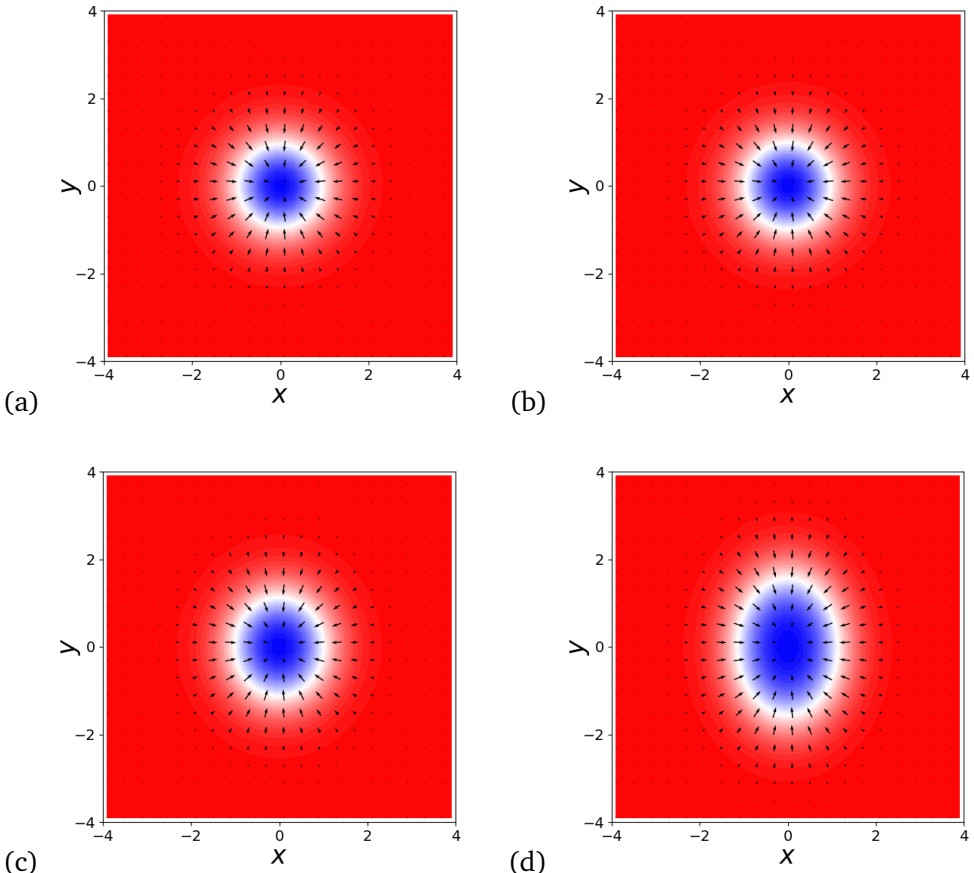

Figure 1: The field $\boldsymbol{n}$ for a static and for traveling skyrmions for the parameter value $\lambda = 0.45$. Entry (a) shows the static (axially symmetric) skyrmion and the remaining entries show traveling skyrmions with velocities (b) $v = 0.2$, (c) $v = 0.4$, and (d) $v = 0.6$. Vectors show the projection of $\boldsymbol{n}$ on the plane and colors denote the $n_3$ component (red means $n_3 > 0$, blue means $n_3 < 0$ and white is for $n_3 \approx 0$). The skyrmions get elongated perpendicular to the direction of propagation ($y$ direction) as the velocity increases. Their size along the direction of propagation ($x$ direction) also increases with the velocity albeit at a much slower rate than in the $y$ direction.

The key to understanding the behavior of the maximum velocity $v_c$ is the numerical finding that the skyrmion expands in both the $x$ and $y$ directions as $v$ approaches $v_c$. In the limit $v \to v_c$, we could try to study the system separately in the two directions. For this purpose, it is helpful to write Eq. (15) in the form

$$\boldsymbol{n} \times \left\{ \left[ (1 - v^2) \partial_1^2 \boldsymbol{n} - \lambda \hat{\boldsymbol{e}}_2 \times \partial_1 \boldsymbol{n} \right] + \left[ \partial_2^2 \boldsymbol{n} + \lambda \hat{\boldsymbol{e}}_1 \times \partial_2 \boldsymbol{n} \right] + n_3 \hat{\boldsymbol{e}}_3 \right\} = 0, \tag{16}$$

grouping together the terms with partial derivatives in the same direction. In the limit $v \to v_c$ where the skyrmion is very elongated in the $y$ direction, we could study the profile on the $x$ axis neglecting the $y$ derivatives. The obtained one-dimensional (1D) equation has a stable uniform state and a domain wall solution when the effective DM parameter is smaller than $2/\pi$. For Eq. (16) this gives

$$\frac{\lambda}{\sqrt{(1 - v^2)}} \leq \frac{2}{\pi} \Rightarrow |v| \leq \sqrt{1 - \frac{\pi^2}{4} \lambda^2} \equiv v_c. \tag{17}$$

Table 1: Values of $v_c$ obtained numerically for various values of the parameter $\lambda$. They are found to be in good agreement with Eq. (17).

| $\lambda$ | $v_c$ |
|-----------|-------|
| 0.35 | 0.85 |
| 0.40 | 0.79 |
| 0.45 | 0.715 |
| 0.50 | 0.63 |
| 0.55 | 0.51 |
| 0.60 | 0.35 |

The results of numerical simulations give, indeed, values for maximum skyrmion velocities to within about 1% of those obtained from formula (17), as can be seen in Table 1. Numerical simulations for skyrmion configurations at values of $\lambda$ close to zero or close to $2/\pi$ are more complicated because of the multiscale character of the solutions at these values of $\lambda$ [26, 27]. The exploration of the entire range of $\lambda$ values numerically would require the development of special numerical methods.

In the above considerations, we viewed the skyrmion domain wall as a 1D domain wall in the skyrmion elongation direction (perpendicular to the $x$ axis). This lead to the successful prediction of the maximum skyrmion velocity $v_c$ by Eq. (17), meaning that this velocity is established by the same mechanism that is responsible for the distabilisation of the uniform to the spiral state.

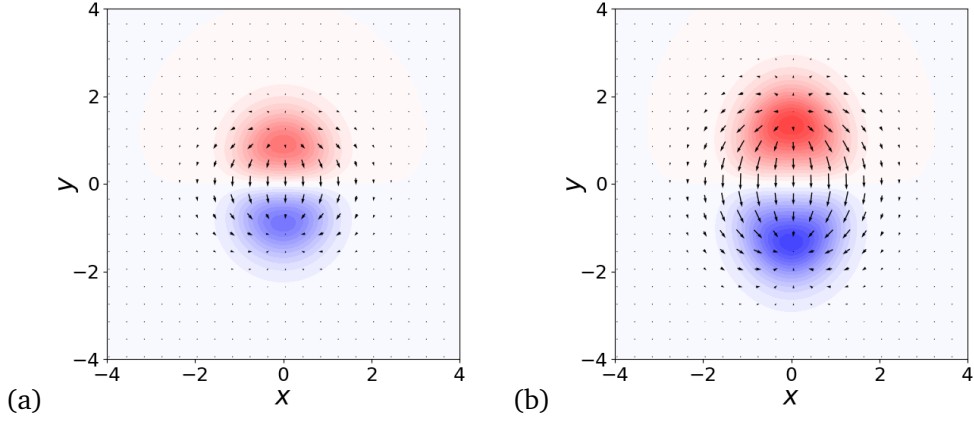

(a)  (b)

Figure 2: The magnetization $\boldsymbol{m}/\epsilon$ for the traveling skyrmions shown in Fig. 1 for velocities (a) $v = 0.4$ and (b) $v = 0.6$. There is a net magnetization in the $\hat{\boldsymbol{e}}_2$ direction and this is increasing with the propagation velocity as indicated in Fig. 3. For the values of the components of $\boldsymbol{m}/\epsilon$, we have (a) $-0.113 < m_2/\epsilon < 0.023$, $|m_3|/\epsilon < 0.077$ and (b) $-0.168 < m_2/\epsilon < 0.034$, $|m_3|/\epsilon < 0.106$. The representation of the vector $\boldsymbol{m}$ follows the conventions explained in Fig. 1.

A significant feature of traveling AFM skyrmions is seen in their magnetization vector given in Eq. (10). For traveling solutions of the form (13) we have

$$\boldsymbol{m} = \frac{\epsilon v}{2\sqrt{2}}\, \partial_1 \boldsymbol{n} \times \boldsymbol{n}. \tag{18}$$

Fig. 2 shows the vector $\boldsymbol{m}/\epsilon$ for traveling skyrmions for two values of the velocity. The magnetization vector is divided by $\epsilon$ for the reasons explained following Eq. (10). The third component

of $\boldsymbol{m}$ has opposite values in the upper and the lower half of the skyrmion. More interesting is the fact that the in-plane component of $\boldsymbol{m}$ points in the negative $y$ axis.

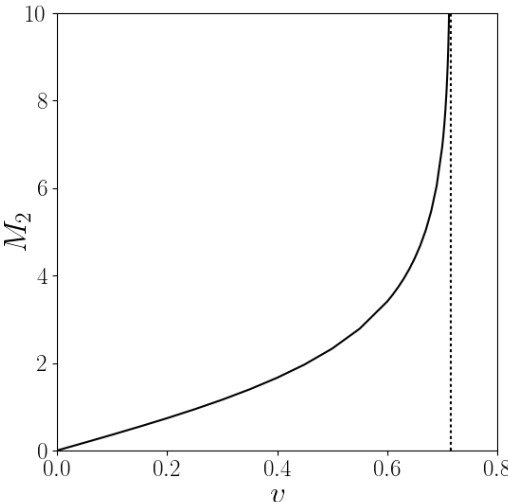

Figure 3: The total magnetization $M_2$ defined in Eq. (19) of a traveling skyrmion as a function of the skyrmion velocity $v$. The parameter value is $\lambda = 0.45$ and the vertical dotted line marks the maximum value of the velocity $v_c \simeq 0.715$.

Let us define the total magnetization along $y$ by

$$M_2 = \frac{1}{\epsilon} \int m_2 \, dx dy = \frac{v}{2\sqrt{2}} \int \hat{\boldsymbol{e}}_2 \cdot (\partial_1 \boldsymbol{n} \times \boldsymbol{n}) \, dx dy. \tag{19}$$

The integrand is the Lifshitz invariant $\mathcal{L}_{21}$ defined in Eq. (9). The integral in Eq. (19) gives a nonzero negative result for a skyrmion as it is proportional to a part of the DM energy. Similarly, for a Néel domain wall perpendicular to the $x$ axis, it gives the result $-\pi$. Consequently, as the velocity increases, and the skyrmion gets elongated in the $y$ direction, the integrated $\mathcal{L}_{21}$ is expected to grow in absolute value proportionally to the skyrmion length along the $y$ axis. This is indeed confirmed by the numerical data to a good approximation for every velocity $v$. Fig. 3 shows $M_2$ as a function of the velocity $v$. It is approximately linear for small velocities due to the factor $v$ in Eq. (19) and it diverges as $v \to v_c$ where the skyrmion becomes infinitely elongated.

Reversing vector $\boldsymbol{m}$ would lead to a skyrmion moving in the opposite direction. Otherwise, skyrmions with negative and positive velocities, $\pm v$, have the same configuration of $\boldsymbol{n}$.

The traveling skyrmion configurations for $\boldsymbol{n}$ shown in Fig. 1 and the associated fields $\boldsymbol{m}, \boldsymbol{k}, \boldsymbol{l}$, obtained by Eqs. (39) and (40), can be used to find the spins at each tetramer via Eq. (43). As a check of consistency we have tested the dynamics of the spin configurations $\boldsymbol{S}_{i,j}$ obtained from the configuration $\boldsymbol{n}$ of traveling skyrmions such as those in Fig. 1. We propagate in time under Eq. (5) a skyrmion configuration in the spin lattice and we verify that this propagates rigidly with a velocity $2\sqrt{2}asJv$, where $v$ is the skyrmion velocity in the $\sigma$-model and $a$ the lattice spacing in the spin lattice. The factor $2\sqrt{2}asJ$ is due to the definition of the scaled time (38) in the $\sigma$-model. This provides a verification of the consistency of the original equations (5) for the spins with the continuum approximation (7).

The detailed description of the configuration for a traveling skyrmion can serve as a guide for setting-up schemes to obtain these in experiments. Engineering the vector $\boldsymbol{m}$ in order to obtain configurations such as those in Fig. 2 could lead to such methods.

## 4 Particle-like character

One of the most interesting features of solitons is that they behave as particles. We will study this making extensive use of the relations derived in App. B. In order to explore the details of its particle-like character, we study the energy (8) of the skyrmion as a function of velocity. For small velocities we may assume that the traveling skyrmion configuration is $n(x, y, \tau; \upsilon) \approx n_0(x - \upsilon\tau, y)$. Then, the energy takes the form

$$E = \frac{1}{2}\int \dot{n}^2 \, dx dy + E_0 = \frac{\upsilon^2}{2}\int (\partial_1 n_0)^2 \, dx dy + E_0 \,, \qquad (20)$$

where $E_0$ is the energy of the static skyrmion. We set

$$\mathcal{M}_0 = \int (\partial_1 n_0)^2 dx dy \qquad (21)$$

and write the energy for small velocities as

$$E(\upsilon) = \frac{1}{2}\mathcal{M}_0 \upsilon^2 + E_0, \qquad \upsilon \ll \upsilon_c. \qquad (22)$$

This has the form of the energy of a Newtonian particle with a rest mass $\mathcal{M}_0$.

A measure of the size of the skyrmion is given by the total number of reversals for the third component of the vector $n$, defined as

$$\mathcal{N} = \int (1 - n_3) dx dy. \qquad (23)$$

In the simple case of a circular region of radius $R$ where $n_3 = -1$, we would have $R = \sqrt{\mathcal{N}/(2\pi)}$.

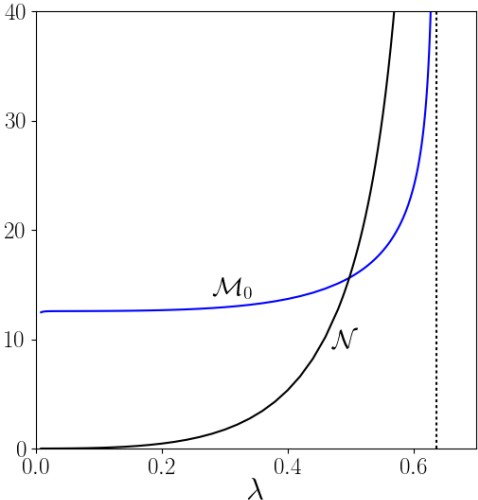

Figure 4: The rest mass $\mathcal{M}_0$ of a skyrmion defined in Eq. (21) and the Néel vector reversals $\mathcal{N}$ defined in Eq. (23) as functions of the dimensionless DM parameter $\lambda$. In the limit $\lambda \to 0$, we have $\mathcal{M}_0 = 4\pi$ and $\mathcal{N} = 0$. The dotted vertical line marks the value $\lambda = 2/\pi$ where the skyrmion radius diverges to infinity.

The mass of the skyrmion as well as $\mathcal{N}$ depend on the skyrmion profile. Changing the DM parameter $\lambda$ changes the skyrmion profile. Fig. 4 shows the rest mass $\mathcal{M}_0$ and the Néel vector

reversals $\mathcal{N}$ for a static skyrmion as functions of the DM parameter. For $\lambda \to 0$ the skyrmion radius goes to zero and the skyrmion profile approaches that of the BP skyrmion [26]. Thus, in this limit, $\mathcal{M}_0 \to 4\pi$ and $\mathcal{N} \to 0$. For $\lambda \to 2/\pi$, the skyrmion radius goes to infinity and the skyrmion profile is described via a domain wall similar to the 1D domain wall [27,30]. In this limit, $\mathcal{M}_0$ is proportional to the skyrmion radius and $\mathcal{N}$ is proportional to the skyrmion area (radius squared) while they both diverge to infinity.

The components of the linear momentum $(P_1, P_2)$ in this system are given by [11,31]

$$P_1 = -\int \dot{\boldsymbol{n}} \cdot \partial_1 \boldsymbol{n}\, dx dy, \quad P_2 = -\int \dot{\boldsymbol{n}} \cdot \partial_2 \boldsymbol{n}\, dx dy. \tag{24}$$

For a traveling wave as in Eq. (13) only the first component $P = P_1$ is nonzero and we have

$$P = \mathcal{M}v, \tag{25}$$

where we have defined the mass

$$\mathcal{M} = \int (\partial_1 \boldsymbol{n})^2\, dx dy \tag{26}$$

that depends on the velocity, $\mathcal{M} = \mathcal{M}(v)$, via the skyrmion configuration. For small velocities the assumption $\boldsymbol{n}(x, y, \tau; v) \approx \boldsymbol{n}_0(x - v\tau, y)$ leads to

$$P = v \int (\partial_1 \boldsymbol{n}_0)^2\, dx dy = \mathcal{M}_0 v, \qquad v \ll v_c. \tag{27}$$

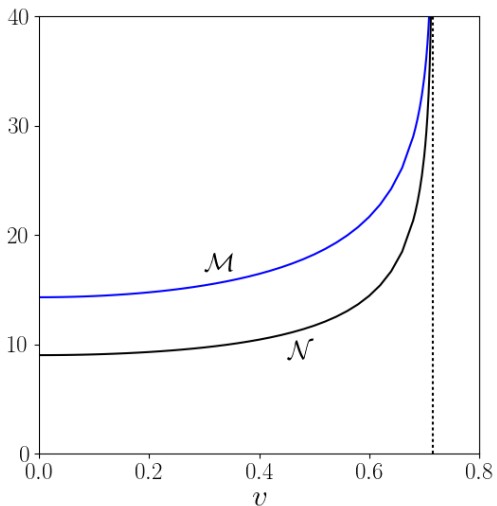

Figure 5: The mass $\mathcal{M}$ defined in Eq. (26), and the Néel vector reversals $\mathcal{N}$ in Eq. (23) of a traveling skyrmion as functions of the skyrmion velocity $v$. The parameter value is $\lambda = 0.45$. The vertical dotted line marks the maximum value of the velocity $v_c \simeq 0.715$.

Fig. 5 shows the numerically obtained mass $\mathcal{M}$ and Néel vector reversals $\mathcal{N}$ as functions of the skyrmion velocity. Both quantities increase as the velocity increases and they diverge to infinity as $v \to v_c$. The key for the understanding of the behavior of $\mathcal{M}$ for velocities close to $v_c$ is the numerical finding that the skyrmion gets elongated in the $y$ direction. Since $\mathcal{M}$ depends

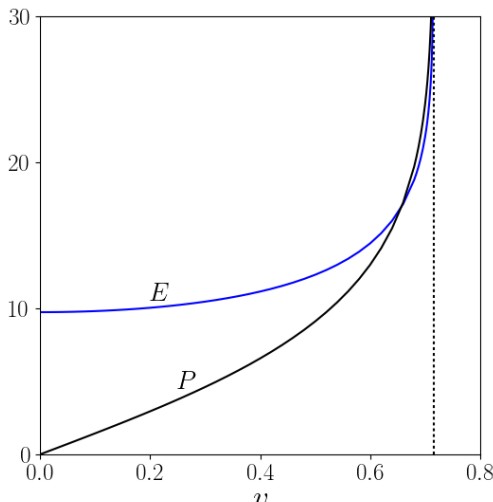

Figure 6: The energy (8) and the linear momentum $P$ given in Eq. (25) for a traveling skyrmion as a function of its velocity $v$, for $\lambda = 0.45$. The behavior at small velocities is given by Eqs. (22) and (27) for $\mathcal{M}_0 = 14.29$, $E_0 = 9.73$. The vertical dotted line marks the maximum value for the velocity $v_c \simeq 0.715$, where both $E$ and $P$ diverge to infinity.

on a derivative along $x$ only, we are lead to the conclusion that it should be proportional to the skyrmion length along the $y$ axis in the case of large elongation. This is indeed verified by the numerical data.

Fig. 6 shows the energy $E$ and the linear momentum $P$ of a traveling skyrmion as functions of the velocity $v$. The numerical results shown in the figure verify the linear dependence of $P$ on the velocity for small $v$ with a proportionality constant equal to $\mathcal{M}_0$. The parabolic form of the energy (22) with the same constant $\mathcal{M}_0$ is also verified. For velocities close to $v_c$ the energy and the linear momentum diverge to infinity thus showing relativistic behavior.

The group velocity relation

$$v = \frac{dE}{dP} \tag{28}$$

is verified by our numerical results for the entire range of linear momenta. The energy-momentum relation for small velocities is obtained from Eqs. (22) and (27),

$$E \approx E_0 + \frac{P^2}{2\mathcal{M}_0}, \qquad v \ll v_c, \tag{29}$$

and it is consistent with Eq. (28). For large momenta we can substitute $v \approx v_c$ in Eq. (28) and obtain

$$E \approx v_c P + E_c, \qquad v \to v_c, \tag{30}$$

where $E_c$ is a constant. Formula (30) fits very well the numerical data for $E_c = 4.5$ for the parameter value $\lambda = 0.45$. Fig. 7 shows the dispersion relation (energy vs momentum) for the numerically calculated traveling skyrmions. The combination of the two forms (29) and (30), that are also plotted in the figure, give an excellent approximation for almost the entire range of linear momenta.

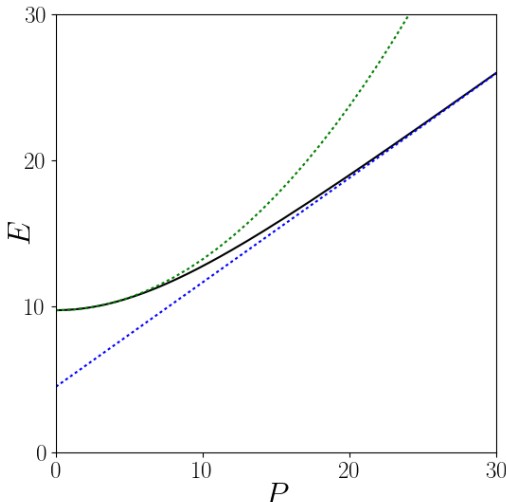

Figure 7: The energy–momentum dispersion for the traveling skyrmions, for $\lambda = 0.45$, is shown by a solid black line. For small momenta the relation is parabolic as shown in Eq. (29) for $\mathcal{M}_0 = 14.29$, $E_0 = 9.73$, that it is plotted by a green dotted line. The dispersion becomes linear for large momenta according to Eq. (30), for $E_c = 4.5$, that is plotted by a blue dotted line.

Virial relations for the traveling skyrmions are derived in Appendix B. Eq. (54) for the energy can be written as

$$E = \mathcal{M} + \lambda \int \hat{\boldsymbol{e}}_2 \cdot (\partial_1 \boldsymbol{n} \times \boldsymbol{n}) dx dy. \tag{31}$$

The second term on the right hand side is the integrated Lifshitz invariant $\mathcal{L}_{21}$. In the case of a static skyrmion, it is proportional to the DM energy and it gives a negative contribution. We thus conclude that the DM interaction modifies the relation between energy and mass compared to standard Lorentz invariant models (see Appendix C). The integrated Lifshitz invariant in Eq. (31) and the mass $\mathcal{M}$ are both proportional to the length of the traveling skyrmion in the $y$ direction as has been discussed in the paragraphs following Eq. (19) and Eq. (26) respectively.

Eq. (31) takes an interesting form if we use Eq. (53). We obtain

$$E = \mathcal{M}v^2 + \int \partial_2 \boldsymbol{n} \cdot \partial_2 \boldsymbol{n} \, dx dy - \lambda \int \hat{\boldsymbol{e}}_1 \cdot (\partial_2 \boldsymbol{n} \times \boldsymbol{n}) dx dy. \tag{32}$$

Since the skyrmion gets strongly elongated in the $y$ direction for large velocities, the two last terms on the right hand side, that contain only $y$ derivatives, become negligible compared to the first term in the limit $v \to v_c$. Therefore, Eq. (32) contains more concrete information than its equivalent Eq. (31). Specifically, we find a very good approximation of the numerical data for large velocities $v \approx v_c$, using the simplified version of Eq. (32)

$$E \approx \mathcal{M}v_c^2 + E_c, \qquad v \to v_c. \tag{33}$$

An alternative way to obtain Eq. (33) is to use Eq. (30) with $P \approx \mathcal{M}v_c$, that is valid for $v \approx v_c$. In that case, Eq. (33) taken in combination with Eq. (32) prove that the energy shift is obtained

as

$$\int \partial_2 \boldsymbol{n} \cdot \partial_2 \boldsymbol{n} \, dx dy - \lambda \int \hat{\boldsymbol{e}}_1 \cdot (\partial_2 \boldsymbol{n} \times \boldsymbol{n}) dx dy \xrightarrow{v \to v_c} E_c. \tag{34}$$

Eq. (33) can be compared with the standard relativistic relation in Eq. (56). The two equations differ in that $v_c < 1$ is smaller that the velocity of light in the Lorentz invariant model and in that the chiral model introduces a constant shift $E_c$ in the energy in the relativistic limit $v \to v_c$.

## 5 Concluding remarks

We have given a detailed description of traveling skyrmions in antiferromagnets with the Dzyaloshinskii-Moriya interaction. The study is based on a nonlinear $\sigma$-model that is derived as the continuum approximation of the original discrete model for a lattice of spins with antiferromagnetic interactions. We first consider the fundamental argument that has been applied within the Landau-Lifshitz equation for a ferromagnet to show that traveling solitary waves are prohibited due to a link between the skyrmion number $Q$ and the dynamics [18]. We then apply, in Appendix B, the corresponding argument in the $\sigma$-model studied here to show that no such link between topology and dynamics exists and traveling solitary waves are allowed within this theory. This result is the main motivation for the present work.

We find numerically the configurations of traveling skyrmion solutions for velocities $|v| < v_c$ where the maximum velocity $v_c$ depends on the dimensionless DM parameter $\lambda$. We observe that traveling skyrmions are elongated perpendicular to the direction of propagation and they apparently get infinitely elongated in the limit $v \to v_c$. We find that a net magnetization is developed with orientation perpendicular to the direction of propagation and we suggest that this could offer a measurable quantity in order to observe propagating AFM skyrmions. We obtain a formula for the maximum skyrmion velocity $v_c$ based on the argument that this is established by the same mechanism that is responsible for the distabilisation of the uniform to the spiral state. The velocity $v_c$ is smaller than unity, i.e., than the maximum velocity within the Lorentz invariant model obtained when the DM interaction is absent.

We define the mass and give the dispersion relation for traveling skyrmions. We derive virial relations and obtain exact and approximate relations between the mass the energy and the linear momentum of skyrmions. These clarify their particle-like features and substantiate their Newtonian and relativistic character for low and for large momenta respectively.

We remind the reader that, in the case of an FM, the topological skyrmion with the standard skyrmion number $Q = 1$ shows Hall dynamics while a non-topological skyrmionium with $Q = 0$ shows Newtonian dynamics [32]. It has thus been clear that the dynamics of solitons in a FM depends crucially on their skyrmion number. In the presently studied AFM, we have seen that the dynamics of a skyrmion (with $Q = 1$) is Newtonian for small velocities. It is thus dramatically different that the dynamics of a $Q = 1$ FM skyrmion. On the other hand, it appears, counterintuitively, to be similar to the dynamics of a FM skyrmionium. The similarities can be seen in the energy and momentum behavior shown in Fig. 6 as well as in the configuration of the traveling AFM skyrmion shown in Fig. 1 when these are compared with the corresponding figures for the FM skyrmionium [32].

The study of dynamics of solitons or other configurations in AFM within suitable $\sigma$-models can give rise to a wealth of dynamical phenomena [33–35] that have not been there in FM studied within the Landau-Lifshitz equation. This opens wide perspectives for the study of AFM dynamics, provided measurable quantities of AFM magnetic order be identified.

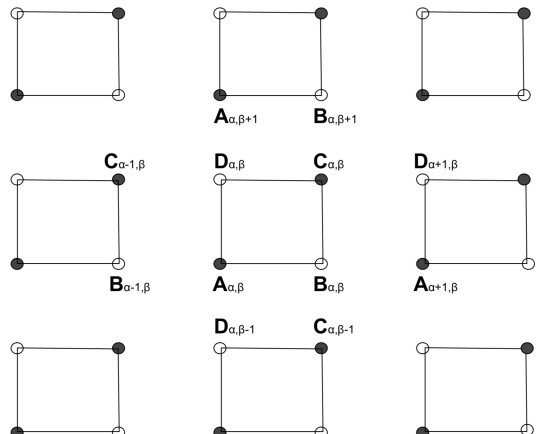

Figure 8: A tetramerization of the square lattice. The tetramers are indexed by integers $\alpha, \beta$ and the spins at each tetramer are denoted by $A, B, C, D$.

# Acknowledgements

SK acknowledges financial support from the Hellenic Foundation for Research and Innovation (HFRI) and the General Secretariat for Research and Technology (GSRT), under grant agreement No 871.

# A   Derivation of the continuum model

For the derivation of a continuum model we first need to define an appropriate order parameter with a continuum limit. In order to do this, we consider a tetramerization of the square lattice as shown in Fig. 8 and each tetramer is labelled by two indices $\alpha$ and $\beta$ numbered consecutively in the horizontal and vertical directions respectively. In Ref. [11], it is shown that the continuum model for the AFM takes a simpler form when it is derived on tetramers instead of dimers. We repeat this derivation in a compact form including now the DM term. Derivations and applications of DM terms in AFM are found in [2, 8, 17, 36, 37].

At each tetramer, we denote the spin values by $A_{\alpha,\beta}, B_{\alpha,\beta}, C_{\alpha,\beta}, D_{\alpha,\beta}$. A convenient set of fields is defined at each tetramer by the linear combinations

$$
\begin{aligned}
m &= \frac{1}{4s}(A + B + C + D) \\
n &= \frac{1}{4s}(A - B + C - D) \\
k &= \frac{1}{4s}(A + B - C - D) \\
l &= \frac{1}{4s}(A - B - C + D).
\end{aligned}
\tag{35}
$$

The vector $m$ gives the normalised magnetization at each tetramer and $n$ is called the Néel vector. The equations for the four spins at each tetramer in Fig. 8 with indices $\alpha, \beta$ are derived

from Eq. (5) and they read

$$
\begin{aligned}
\frac{\partial A_{\alpha,\beta}}{\partial t} &= A_{\alpha,\beta} \times \left\{ -J(B_{\alpha,\beta} + B_{\alpha-1,\beta} + D_{\alpha,\beta} + D_{\alpha,\beta-1}) \right. \\
&\quad \left. + D[\hat{e}_2 \times (B_{\alpha,\beta} - B_{\alpha-1,\beta}) - \hat{e}_1 \times (D_{\alpha,\beta} - D_{\alpha,\beta-1})] + g A_{\alpha,\beta} \cdot \hat{e}_3 \right\} \\
\frac{\partial B_{\alpha,\beta}}{\partial t} &= B_{\alpha,\beta} \times \left\{ -J(A_{\alpha,\beta} + A_{\alpha+1,\beta} + C_{\alpha,\beta} + C_{\alpha,\beta-1}) \right. \\
&\quad \left. + D[\hat{e}_2 \times (A_{\alpha+1,\beta} - A_{\alpha,\beta}) - \hat{e}_1 \times (C_{\alpha,\beta} - C_{\alpha,\beta-1})] + g B_{\alpha,\beta} \cdot \hat{e}_3 \right\} \\
\frac{\partial C_{\alpha,\beta}}{\partial t} &= C_{\alpha,\beta} \times \left\{ -J(D_{\alpha,\beta} + D_{\alpha+1,\beta} + B_{\alpha,\beta} + B_{\alpha,\beta+1}) \right. \\
&\quad \left. + D[\hat{e}_2 \times (D_{\alpha+1,\beta} - D_{\alpha,\beta}) - \hat{e}_1 \times (B_{\alpha,\beta+1} - B_{\alpha,\beta})] + g C_{\alpha,\beta} \cdot \hat{e}_3 \right\} \\
\frac{\partial D_{\alpha,\beta}}{\partial t} &= D_{\alpha,\beta} \times \left\{ -J(C_{\alpha,\beta} + C_{\alpha-1,\beta} + A_{\alpha,\beta} + A_{\alpha,\beta+1}) \right. \\
&\quad \left. + D[\hat{e}_2 \times (C_{\alpha,\beta} - C_{\alpha-1,\beta}) - \hat{e}_1 \times (A_{\alpha,\beta+1} - A_{\alpha,\beta})] + g D_{\alpha,\beta} \cdot \hat{e}_3 \right\}.
\end{aligned}
\tag{36}
$$

We consider a small parameter $\epsilon$ in terms of which the cartesian coordinates are

$$
x = 2\epsilon(\alpha - \alpha_0), \qquad y = 2\epsilon(\beta - \beta_0),
\tag{37}
$$

where $\alpha_0, \beta_0$ are constants defining the central point of the lattice of tetramers. As $\epsilon \to 0$ the coordinates in Eq. (37) become continuous variables. In the same limit, we assume that the fields $A_{\alpha,\beta}, B_{\alpha,\beta}, C_{\alpha,\beta}, D_{\alpha,\beta}$ and also those in Eq. (35) approach continuous limits and we use the relations

$$
\begin{aligned}
A_{\alpha\pm1,\beta} &= A \pm 2\epsilon \partial_1 A + 2\epsilon^2 \partial_1^2 A, \\
A_{\alpha,\beta\pm1} &= A \pm 2\epsilon \partial_2 A + 2\epsilon^2 \partial_2^2 A,
\end{aligned}
$$

and similar relations for the fields $B, C$ and $D$. The notation $\partial_1, \partial_2$ denotes differentiation with respect to $x, y$ respectively. Taking appropriate combinations of Eqs. (36) we derive dynamical equations for the fields (35). For the set of equations to be consistent in the various orders of $\epsilon$, we assume that $n \sim O(1)$ and $m, k, l \sim O(\epsilon)$. From the definitions in Eq. (35) we find that $m \cdot n = k \cdot n = l \cdot n = 0$ in the limit $\epsilon \to 0$. We further rescale time according to

$$
\tau = 2\sqrt{2}\epsilon s J t.
\tag{38}
$$

In the equations for $k, l$, the time derivative does not enter in the order $O(\epsilon)$ and we obtain

$$
k = -\frac{\epsilon}{2} \partial_1 n, \qquad l = -\frac{\epsilon}{2} \partial_2 n.
\tag{39}
$$

The equation for $n$ gives, to $O(\epsilon)$,

$$
\epsilon \dot{n} = 2\sqrt{2} m \times n
$$

and this is solved for $m$ to give

$$
m = \frac{\epsilon}{2\sqrt{2}} (n \times \dot{n}),
\tag{40}
$$

where the dot denotes differentiation with respect to the rescaled time $\tau$.

Finally, the dynamical equation for $m$ gives, in the order $O(\epsilon^2)$,

$$
2\sqrt{2}\epsilon \dot{m} = \epsilon^2 n \times (\partial_1^2 n + \partial_2^2 n) + \epsilon \frac{2D}{J} \epsilon_{\mu\nu} n \times (\hat{e}_\mu \times \partial_\nu n) + \frac{g}{J} n \times n_3 \hat{e}_3,
$$

where we have used Eqs. (39). We introduce the rescaled parameters $\kappa, \lambda$ defined from

$$g = \epsilon^2 J \kappa, \qquad D = \epsilon J \lambda, \tag{41}$$

and insert $\boldsymbol{m}$ from Eq. (40) to obtain the equation for the Néel vector,

$$\boldsymbol{n} \times \left( \ddot{\boldsymbol{n}} - \Delta \boldsymbol{n} - 2\lambda \epsilon_{\mu\nu} \hat{\boldsymbol{e}}_{\mu} \times \partial_{\nu}\boldsymbol{n} - \kappa n_3 \hat{\boldsymbol{e}}_3 \right) = 0. \tag{42}$$

The process of finding an actual AFM configuration proceeds as follows. One first solves Eq. (42) and then the fields in Eqs. (39), (40) are calculated. Finally, relations (35) are inverted to give the spins at each tetramer

$$\begin{aligned} \boldsymbol{A} &= s(\boldsymbol{m} + \boldsymbol{n} + \boldsymbol{k} + \boldsymbol{l}), \quad \boldsymbol{B} = s(\boldsymbol{m} - \boldsymbol{n} + \boldsymbol{k} - \boldsymbol{l}) \\ \boldsymbol{C} &= s(\boldsymbol{m} + \boldsymbol{n} - \boldsymbol{k} - \boldsymbol{l}), \quad \boldsymbol{D} = s(\boldsymbol{m} - \boldsymbol{n} - \boldsymbol{k} + \boldsymbol{l}). \end{aligned} \tag{43}$$

## B Virial relations

Eq. (15) may be written as

$$\boldsymbol{n} \times \boldsymbol{f} = v^2 \boldsymbol{n} \times \partial_1^2 \boldsymbol{n}, \qquad \boldsymbol{f} = -\frac{\delta V}{\delta \boldsymbol{n}}. \tag{44}$$

Following a standard procedure [32] we take the cross product of the latter with $\partial_{\nu}\boldsymbol{n}$ and then the dot product with $\boldsymbol{n}$ to obtain

$$\boldsymbol{f} \cdot \partial_{\nu}\boldsymbol{n} = v^2 \partial_1^2 \boldsymbol{n} \cdot \partial_{\nu}\boldsymbol{n}, \qquad \nu = 1, 2. \tag{45}$$

We write $-\boldsymbol{f} \cdot \partial_{\nu}\boldsymbol{n} = \partial_{\lambda}\sigma_{\nu\lambda}$ [18], where the components of the tensor $\sigma$ are

$$\begin{aligned} \sigma_{11} &= -\frac{1}{2}\partial_1\boldsymbol{n} \cdot \partial_1\boldsymbol{n} + \frac{1}{2}\partial_2\boldsymbol{n} \cdot \partial_2\boldsymbol{n} + \frac{\kappa}{2}(1 - n_3^2) - \lambda \hat{\boldsymbol{e}}_1 \cdot (\partial_2\boldsymbol{n} \times \boldsymbol{n}) \\ \sigma_{12} &= -\partial_1\boldsymbol{n} \cdot \partial_2\boldsymbol{n} + \lambda \hat{\boldsymbol{e}}_1 \cdot (\partial_1\boldsymbol{n} \times \boldsymbol{n}) \\ \sigma_{21} &= -\partial_1\boldsymbol{n} \cdot \partial_2\boldsymbol{n} - \lambda \hat{\boldsymbol{e}}_2 \cdot (\partial_2\boldsymbol{n} \times \boldsymbol{n}) \\ \sigma_{22} &= \frac{1}{2}\partial_1\boldsymbol{n} \cdot \partial_1\boldsymbol{n} - \frac{1}{2}\partial_2\boldsymbol{n} \cdot \partial_2\boldsymbol{n} + \frac{\kappa}{2}(1 - n_3^2) + \lambda \hat{\boldsymbol{e}}_2 \cdot (\partial_1\boldsymbol{n} \times \boldsymbol{n}). \end{aligned} \tag{46}$$

We further note that

$$\begin{aligned} \partial_1^2\boldsymbol{n} \cdot \partial_1\boldsymbol{n} &= \partial_1\left(\frac{1}{2}\partial_1\boldsymbol{n} \cdot \partial_1\boldsymbol{n}\right) \\ \partial_1^2\boldsymbol{n} \cdot \partial_2\boldsymbol{n} &= \partial_1(\partial_1\boldsymbol{n} \cdot \partial_2\boldsymbol{n}) + \partial_2\left(-\frac{1}{2}\partial_1\boldsymbol{n} \cdot \partial_1\boldsymbol{n}\right) \end{aligned} \tag{47}$$

and Eq. (45) gives the convenient forms

$$\begin{aligned} \partial_{\lambda}\sigma_{1\lambda} &= v^2 \partial_1\left(-\frac{1}{2}\partial_1\boldsymbol{n} \cdot \partial_1\boldsymbol{n}\right) \\ \partial_{\lambda}\sigma_{2\lambda} &= v^2\left[\partial_1(-\partial_1\boldsymbol{n} \cdot \partial_2\boldsymbol{n}) + \partial_2\left(\frac{1}{2}\partial_1\boldsymbol{n} \cdot \partial_1\boldsymbol{n}\right)\right]. \end{aligned} \tag{48}$$

Solutions representing configurations that propagate with velocity $v$ and satisfy Eq. (15) or (44), also satisfy Eq. (48). As both sides in Eqs. (48) are total derivatives, they both vanish if one takes their integral over the entire plane and a trivial identity is obtained for any velocity $v$. The significance of this result is revealed when it is compared with the corresponding

calculation for the Landau-Lifshitz equation in a FM. In the latter case, one obtains the relation (see Eq. (4.5) in Ref. [18])

$$vQ = 0 \qquad \text{[in FM]}, \tag{49}$$

which contains the skyrmion number $Q$ and it is satisfied only for $v = 0$ when $Q \neq 0$. As a result, traveling solitary waves with $Q \neq 0$ do not exist in a FM. As the possibility of topological solitons traveling with $v \neq 0$ is *not* excluded by the corresponding analysis for a AFM, this constitutes the fundamental difference between the dynamics in a AFM compared to a FM. Stated in a more general way, the skyrmion number $Q$ is unrelated to the dynamics in AFMs, in stark contrast to the link between topology and dynamics in FMs. The calculations presented in this paper have been motivated by Eqs. (48) and the consequences stated above.

We take moments of Eqs. (48) with $x_\nu$ for $\nu = 1, 2$, integrate both sides over the entire plane and apply the divergence theorem [38] to obtain four independent virial relations that must be satisfied by any traveling solution,

$$
\begin{aligned}
\int \sigma_{11}\, dx dy &= -\frac{v^2}{2} \int \partial_1 \boldsymbol{n} \cdot \partial_1 \boldsymbol{n}\, dx dy \\
\int \sigma_{12}\, dx dy &= 0 \\
\int \sigma_{21}\, dx dy &= -v^2 \int \partial_1 \boldsymbol{n} \cdot \partial_2 \boldsymbol{n}\, dx dy \\
\int \sigma_{22}\, dx dy &= \frac{v^2}{2} \int \partial_1 \boldsymbol{n} \cdot \partial_1 \boldsymbol{n}\, dx dy.
\end{aligned}
\tag{50}
$$

Combinations of the above give convenient virial relations. First, we take the special combination

$$\int (\sigma_{11} + \sigma_{22}) dx dy = 0 \Rightarrow E_{\text{DM}} + 2 E_{\text{a}} = 0. \tag{51}$$

This is identical to the virial relation that can be obtained for static skyrmions through Derrick's scaling argument [39] (see also Refs. [32,40]). Thus, Eq. (51) is satisfied by all static as well as traveling skyrmion solutions presented in this paper.

A second virial relation is obtained if we take the combination $\sigma_{12} - \sigma_{21}$, that gives

$$v^2 \int \partial_1 \boldsymbol{n} \cdot \partial_2 \boldsymbol{n}\, dx dy = \lambda \int \hat{\boldsymbol{e}}_\mu \cdot (\partial_\mu \boldsymbol{n} \times \boldsymbol{n}) dx dy. \tag{52}$$

The term on the right hand side coincides with the formula for the bulk DM energy. We find numerically that both sides in the above relation are zero for every velocity $v$. This is due to the parity symmetries that are apparent in all entries of Fig. 1 for the skyrmion configurations.

A third virial relation is obtained by the combination $\sigma_{11} - \sigma_{22}$, that gives

$$\int \left[ \partial_2 \boldsymbol{n} \cdot \partial_2 \boldsymbol{n} - (1 - v^2) \partial_1 \boldsymbol{n} \cdot \partial_1 \boldsymbol{n} \right] dx dy = \lambda \int \left[ \hat{\boldsymbol{e}}_1 \cdot (\partial_2 \boldsymbol{n} \times \boldsymbol{n}) + \hat{\boldsymbol{e}}_2 \cdot (\partial_1 \boldsymbol{n} \times \boldsymbol{n}) \right] dx dy. \tag{53}$$

For the skyrmions in this paper (of Néel type) the first term on the right hand side is positive and the second term is negative.

A useful relation for the energy is obtained if we subtract $\int \sigma_{11} dx dy$ from the energy and use the first of Eqs. (50),

$$E = \int (\partial_1 \boldsymbol{n})^2 dx dy + \lambda \int \hat{\boldsymbol{e}}_2 \cdot (\partial_1 \boldsymbol{n} \times \boldsymbol{n}) dx dy. \tag{54}$$

The first term on the right hand side gives the mass of the traveling skyrmion defined in Eq. (26).

We have verified that all virial relations presented in this Appendix are satisfied by the numerically calculated traveling skyrmion solutions.

## C  The Lorentz invariant model

We give some results for the Lorentz invariant model, e.g., the one obtained if we omit the DM interaction in the model (7) (see also Ref. [31]). The following relations should be compared with those obtained in Appendix B. Let $n_0(x, y)$ be a static solution of the Lorentz invariant model ($\lambda = 0$) and

$$n(x, y, \tau; v) = n_0(\xi, y), \quad \xi = \frac{x - v\tau}{\sqrt{1 - v^2}} = \gamma(x - v\tau)$$

a soliton traveling with velocity $v$, where $\gamma = 1/\sqrt{1 - v^2}$. Eq. (26) for the soliton mass gives

$$\mathcal{M} = \gamma \int (\partial_1 n_0)^2 dx dy = \gamma \mathcal{M}_0, \tag{55}$$

where $\mathcal{M}_0$ is the rest mass. Eq. (54), for $\lambda = 0$, gives

$$E = \mathcal{M}. \tag{56}$$

Eqs. (55), (56) together with Eq. (25) for the linear momentum give the well-known relativistic expression for the energy

$$E^2 = \mathcal{M}_0^2 + P^2. \tag{57}$$

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
