# Peer review of "Traveling skyrmions in chiral antiferromagnets"

_SciPost Physics, doi:SciPost Phys. 8, 086 (2020)_

## Round 1 · Referee Report · Anonymous (Referee 1) · 2020-2-11

Strengths

  1. The paper contains a number of results important for the understanding of the dynamics of antiferromagnetic (AFM) skyrmion.
  2. The paper is well written and is easy to read.

Weaknesses

  1. Most results are obtained by means of numerical calculations. It would be nice to get some analytical estimations, e.g. dependence of skyrmion mass on velocity.
  2. Previous works on AFM skyrmions are not properly cited.

Report

This is a timely and well-written paper which provides a number of results necessary for the proper understanding of the antiferromagnetic AFM skyrmion dynamics: (i) AFM skyrmion demonstrates the inertial motion (without external driving), (ii) the moving AFM skyrmion gains elliptical deformation and (iii) it has properties of a relativistic particle: there exists the maximal velocity and skyrmion mass diverges when reaching the critical velocity.

I suggest the publication with minor revision.

Requested changes

  1. The statement about "the vast difference between the dynamical sectors of the $\sigma$ model and the Landau-Lifshitz equation" made at the end of Section II is at least unclear. The main dynamical equation (9) of the proposed "$\sigma$ model" can be directly obtained from the set of two Landau-Lifshitz equations written for each of the sublattices in the so-called exchange approximation. It was demonstrated (for the first time) in [I], well explained in [II, III] and used for AFM skyrmion description in [IV, V].

  2. AFM skyrmions in the material K$_2$V$_3$O$_8$ were studied in [IV] three years earlier than in [14]. Thus, the paper [IV] should be cited at least together with [14].

  3. Elliptical deformation of the moving AFM skyrmion was first predicted in [VI]. This paper must be cited.

  4. The expression (11) for the magnetization is misleading. It looks like the magnetization vanishes in the continuos limit when $\epsilon \to 0$. On the other hand, the expression for $\mathbf{m}$ in (11) is the so-called dynamical magnetization which generally does not vanish [I, II]. The point is that $\epsilon$ appears in the magnetization (11) because of the time renormalization (A2). Thus, the dynamical magnetization is finite what is consistent with the previous findings. I believe it should be explained after formula (11).

  5. Fig. 5 is discussed in the text earlier than Fig. 4 (page 5).

  6. The large-radius AFM skyrmion was recently considered [VII] within the model of the circularly closed domain wall. Thus [VII] should be cited together with [19,20] on page 5.

[I] I. Bar’yakhtar and B. Ivanov, Sov. J. Low Temp. Phys. 5, 361 (1979). [II] O. Gomonay, V. Loktev, Phys. Rev. B, 81, 144427 (2010). [III] E. Tveten, A. Qaiumzadeh, O. Tretiakov, et al., Phys. Rev. Lett., 110, 127208 (2013). [IV] A. Bogdanov, U. Rößler, M. Wolf, et al., Phys. Rev. B, 66, 214410 (2002). [V] H. Velkov, O. Gomonay, M. Beens, et al., New J. Phys. 18, 075016 (2016). [VI] C. Jin, C. Song, J. Wang, et al., APL 109, 182404 (2016). [VII] V. Kravchuk, O. Gomonay, D. Sheka, et al., Phys. Rev. B 99, 184429 (2019).

---

## Round 1 · Referee Report · Anonymous (Referee 2) · 2020-2-13

Strengths

  1. The paper contains new original results which help to understand the dynamic properties of antiferromagnetic skyrmions.

Weaknesses

  1. It would be nice to have a comparison of numerical calculation with spin lattice simulations.
  2. Almost all results are obtained by means of numerical calculations without any analytical predictions.

Report

The problem of magnetic skyrmion dynamics is interesting and well worth discussing in leading journals.

In the current MS authors study by means of numerical calculations the dynamics of antiferromagnetic (AFM) skyrmions. The authors show that AFM skyrmion behaves as a particle and it has relativistic properties. Authors also show that AFM skyrmions can move without any external driving. These results are interesting and deserve to be published.

I suggest the publication with minor revision.

Requested changes

  1. In the Sec. II.A after Eq. (2) it is mentioned that authors consider material $K_{2}V_{4}O_{8}$ from Ref. [14]. However in Ref. [14] it is considered material $K_{2}V_{3}O_{8}$. Please fix it. Additionally, the AFM skyrmion in such material were studied in [A] earlier than in Ref. [14].

  2. $K_{2}V_{3}O_{8}$ is predicted to have the weak ferromagnetism [B]. Fig. 3 shows that the traveling skyrmion has non-zero magnetization $\vec{m}$. Does AFM skyrmion have its non zero magnetization $\vec{m}$ in statics in your calculations?

  3. It would be nice to have the dependence $v_c(\lambda)$ in Fig.4.

  4. In the second paragraph of Sec. I please fix "LandauLifshitz"$\to$"Landau-Lifshitz".

  5. Fig.5 discussed in the Sec. II.B earlier then Fig. 4. Please fix it.

[A] A. N. Bogdanov, U. K. R{\"o}{\ss}ler, M. Wolf, and K.-H. M{\"u}ller, PRB 66, 214410 (2002) [B] M. D. Lumsden, B. C. Sales, D. Mandrus, S. E. Nagler, and J. R. Thompson, PRL 86, 159 (2001)

---

## Round 2 · Referee Report · Anonymous (Referee 1) · 2020-5-3

Report

Basically, I am satisfied with the revised version. I believe that the elliptical deformation of the moving AFM skyrmion is a nice result, which deserves publication in SciPost.
I have only one objection: in my opinion, the statement
"The type of dynamics of magnetic solitons supported by the σ-model allows for traveling solitons and it is thus very different than the dynamics within the Landau-Lifshitz equation. "
is deeply misleading.
The dynamics of antiferromagnets is described by means the Landau-Lifshitz equations, as well as the dynamics of ferromagnets. For instance, the collinear antiferromagnet is described by a set of two Landau-Lifshitz equations: one equation for each of the sublattices M1 and M2. Each equation is of the first order in time, of course. Then introducing the Neel vector n=(M1-M2)/2 and magnetization vector m=(M1+M2)/2 and assuming that |m|<<|n| one can exclude vector m and obtain the second-order (in time) equation for n. For zero external magnetic field, this equation coincides with (7). This is a very well known procedure, which is widely used in AFM community. For the first time, it was proposed in [10]. The authors just presented an alternative derivation of the previously obtained equation of motion for the Neel vector.

---

## Round 2 · Referee Report · Anonymous (Referee 2) · 2020-5-6

Strengths

The paper contains new original results which help to understand the dynamic properties of antiferromagnetic skyrmions.

Report

The new version of the manuscript successfully addressed the concerns of the referee reports. I recommend its publication in SciPost.

---

## Round 2 · Author Response

We thank the Referees for reading carefully our paper and making a number of useful and constructive comments. We have taken all comments into account in the revised version as explained in our replies to the Referees and the Editor. Furthermore, motivated by their comments we have expanded the paper in significant directions (noted in our response below and in the List of Changes). We give a response to the comments of the Referees as well as a response to the Editor.

Response to Referee 1.

Requested changes.

  1. We refer here to the fact that the Landau-Lifshitz equation for a ferromagnet contains a first time derivative while the sigma model contains a second time derivative. In our paper, we consistently consider that the Landau-Lifshitz equation contains a Laplacian just like the sigma model, while they differ in the time derivative term. Thus, we use the term Landau-Lifshitz equation only for a FM (this is made more explicit in the revised version). In Appendix B, we have added an explicit explanation of the difference between the dynamics in FM and AFM, based on Eqs. (B5) and (B6). We thank the Referee for giving a complete list of references on the subject, which we have included in the revised version.

  2. We have added the suggested paper [IV] as well as the paper reporting the original experimental study for K2V3O8.

  3. We have added paper [VI] and an additional recent one reporting elliptical deformation of skyrmion accelerated by spin torque.

  4. We have added an explanation along the lines suggested by the Referee, following Eq. (10) (in the revised version) for the magnetization. We have also added Fig. 3 for the total magnetization.

  5. Figures have been rearranged. Fig. 3 has been added.

  6. We discuss the divergence of the skyrmion radius at a critical value of the DM parameter as a peripheral issue in connection with Fig. 4. We have cited Ref. [Rohart Thiaville, PRB] as it was the first to quantitatively discuss the divergence of the skyrmion radius. We also cite Ref. [Komineas, Melcher, Venakides, arXiv1910.04818] as it gives a mathematically precise description of a large-radius skyrmion. Several other papers use the idea of a closed domain wall, but as this is not within the scope of this paper, we do not find it necessary to cite these works.

Weaknesses.

  1. We give a list of the analytical results at the end of this reply. Regarding the specific suggestion of the Referee, the dependence of some quantities (mass, integrated Lifshitz invariants) as functions of the skyrmion size is now discussed.

  2. We have now included all relevant references that we know of or that have been brought to our attention.

Response to Referee 2.

Requested changes.

  1. We have fixed the typo. We have added the suggested paper [A] and also paper [B] (reporting the original experimental study for K2V3O8).

  2. We use a model with DM interaction, but this does not include the term responsible for weak ferromagnetism. This was mentioned before Eq. (3) and it is made more explicit in the revised version. Our static skyrmion has zero magnetization vector.

  3. We have derived analytically the dependence of the maximum velocity on the DM parameter. This is verified numerically and the numerical results for various values of the parameter are given in a Table.

  4. Fixed.

  5. Figures have been rearranged.

Weaknesses.

  1. Simulations on the original antiferromagnetic lattice have been performed and are reported in Sec. 3 in the paragraph before the last. An extended report is not necessary as the results are almost identical to the results from the sigma model.

  2. We give a list of the analytical results at the end of this reply.

In response to the comments of the Editor and the Referees we list the mathematical and analytical results now contained in the paper.

  1. Sec. 3. A relaxation algorithm for obtaining traveling wave solutions of the sigma model.
  2. Sec. 3. Dependence of the maximum velocity on the DM parameter.
  3. Sec. 4. Energy and momentum of traveling skyrmions for small velocities based on standard field theory.
  4. Sec. 4. Complete analysis of energy-momentum dispersion (motivated by numerical observations).
  5. Sec. 4. Energy-mass relation given by two different formulas (derived from virial relations). Form of energy-mass for large momenta (as modified by the chiral interaction).
  6. Appendix B. Difference between the dynamics in FM and AFM based on the Hamiltonian structure of the respective theories.
  7. Appendix B. Exact virial relations that are satisfied by traveling solitons. See Eqs. (B8), (B9), (B10), (B11). We note that Appendix B contains results which are fundamental for currying out the present work.

---

## Round 2 · List of Changes

List of changes.

Sec. 1. Added references [Baryakhtar, Ivanov Soviet JLTP 1979], [Bogdanov, Roessler PRB 2002], [Gomonay, Loktev PRB], [Tveten et al PRL], [Velkov et al NJP], [Chmiel et al Nature Materials 2018].

Sec. 2.1, par. 1st. Before Eq. (3), (i) K2V4O8 corrected to K2V3O8, (ii) note about weak ferromagnetism added, (iii) two references added.

Sec. 2.2. Figure and discussion of tetramerization moved to Appendix A.

Sec. 2.2. Parameter kappa eliminated through scaling.

Sec. 2.2. Added Eq. (9) to define Lifshitz invariants, and related discussion.

Sec. 2.2. Added discussion on finiteness of magnetization following Eq. (10).

Sec. 2.2. Last sentences referring to difference between Landau-Lifshitz and sigma model made more precise.

Fig. 1. Last sentence added in caption.

Sec. 3, end of 4th par. Sentence containing two new references added, [Jin et al APL 2016, Salimath et al, PRB 2020].

Sec. 3. Added Eqs. (16), (17) and two paragraphs discussion on the maximum velocity as a function of DM parameter. Added Table 1.

Sec. 3. Added definition of total magnetization, Eq. (19), and discussion following this. Added Figure 3.

Sec. 4. Equation of exchange energy at zero velocity eliminated.

Sec. 4 (previously subsection 3.2). Added paragraph after Eq. (27) about mass vs velocity.

Sec. 4. Discussion of group velocity relation (28) is largely rearranged.

Sec. 4. Discussion of Eq. (31) expanded with more information.

Sec. 4. Eq. (33) is now discussed more explicitly in connection with the group velocity relation, and Eq. (34) added. Current last paragraph in the section is added (the previous last paragraph moved elsewhere).

Sec 5. First and second paragraphs modified.

Acknowledgements added.

App. A. Figure of tetramers moved here from main text and first paragraph modified accordingly.

Appendix B. Added a paragraph containing Eq. (B6).

Appendix C (previously Appendix B.1). It has been shortened.

The wording was changed in various expressions throughout the text.

---

## Round 3 · Author Response

In this resubmission, we respond to the last remark of a Referee and we have corrected typos.

---

## Round 3 · List of Changes

Introduction, 2nd paragraph.
The sentence "The type of dynamics of magnetic solitons supported by the sigma-model allows for traveling solitons and it is thus very different than the dynamics within the Landau-Lifshitz equation."
was substituted for
"The type of dynamics of magnetic solitons supported by the sigma-model allows for traveling solitons and it is thus very different than the dynamics in ferromagnets."

After Eq. (15), we added the following sentence.
Dirichlet boundary conditions are applied with n=(0,0,1) at the lattice end points.

In Eq. (41)/(A7) an extra "s" was erased. It now reads D = epsilon J lambda.
Also, in the unnumbered eqn before Eq. (41)/(A7), a corresponding "s" in the factor of the DM term was a typo and it was erased.

Refs. [26,27] were updated.

---

## Editorial Decision

published